# Cisplatin-Induced Kidney Toxicity: Potential Roles of Major NAD^+^-Dependent Enzymes and Plant-Derived Natural Products

**DOI:** 10.3390/biom12081078

**Published:** 2022-08-05

**Authors:** Amany Iskander, Liang-Jun Yan

**Affiliations:** Department of Pharmaceutical Sciences, College of Pharmacy, University of North Texas Health Science Center, Fort Worth, TX 76107, USA

**Keywords:** cisplatin, kidney toxicity, redox imbalance, mitochondria, natural products, oxidative stress

## Abstract

Cisplatin is an FDA approved anti-cancer drug that is widely used for the treatment of a variety of solid tumors. However, the severe adverse effects of cisplatin, particularly kidney toxicity, restrict its clinical and medication applications. The major mechanisms of cisplatin-induced renal toxicity involve oxidative stress, inflammation, and renal fibrosis, which are covered in this short review. In particular, we review the underlying mechanisms of cisplatin kidney injury in the context of NAD^+^-dependent redox enzymes including mitochondrial complex I, NAD kinase, CD38, sirtuins, poly-ADP ribosylase polymerase, and nicotinamide nucleotide transhydrogenase (NNT) and their potential contributing roles in the amelioration of cisplatin-induced kidney injury conferred by natural products derived from plants. We also cover general procedures used to create animal models of cisplatin-induced kidney injury involving mice and rats. We highlight the fact that more studies will be needed to dissect the role of each NAD^+^-dependent redox enzyme and its involvement in modulating cisplatin-induced kidney injury, in conjunction with intensive research in NAD^+^ redox biology and the protective effects of natural products against cisplatin-induced kidney injury.

## 1. Introduction

Cisplatin is a widely used anti-solid tumor drug that can target a variety of cancers including those of the breast, ovary, lung, testis, head, and neck [1,2,3,4,5]. However, cisplatin’s clinical application and efficacy is highly limited due to its severe adverse effects, in particular, its nephrotoxicity [6,7,8,9,10,11]. It has been estimated that nearly 30% of cancer patients receiving cisplatin treatment could exhibit acute kidney injury (AKI) after the ingestion of a single high dose of cisplatin [12]. If cisplatin-induced AKI is left unmanaged, patients can develop chronic kidney disease (CKD) that can progress to end-stage kidney failure and may also increase the risk of death [13,14,15].

As a vital organ responsible for removal and elimination of cisplatin and its metabolites from the body, the kidney can sustain major damaging effects of cisplatin [16,17]. It is believed that cisplatin accumulates in the proximal tubular region of a nephron [18] (Figure 1) and the proximal tubular epithelial mitochondria are major intracellular sites of cisplatin accumulation [19,20]. Therefore, cisplatin can disrupt mitochondrial function including mitochondrial membrane potential, electron transport chain, the Krebs cycle, and oxidative phosphorylation [20]. Moreover, as the major underlying mechanism of action of cisplatin is its binding to DNA, thus interfering with DNA replication and cancer cell survival, mitochondrial DNA replication in the nephron can also be impaired [20,21], leading to abnormal mitochondrial genesis and kidney dysfunction [21]. Given that the disruption of numerous mitochondrial pathways can eventually converge on the increased mitochondrial generation of reactive oxygen species (ROS), antioxidants, in particular those derived from natural plants, have been widely used to counteract cisplatin kidney toxicity in preclinical and clinical settings [22,23].

In this article, we review the major existing mechanisms of cisplatin-induced kidney injury, the role of major NAD^+^-dependent redox enzymes in cisplatin-induced nephrotoxicity, and the protective effects of natural products that are derived from plants, with particular reference to their anti-oxidative stress roles and redox maintenance capacity. We also briefly cover rodent models used for studying cisplatin-induced kidney injury and highlight the need for dissecting the role of redox-dependent enzymes in treating cisplatin-induced kidney injury.

## 2. Methods

We conducted searches using PubMed, Google, and Science Direct. Search terms included “cisplatin”, “kidney injury” or “renal toxicity” and “nephroprotection” or “nephroprotective”. Among the search results, we only chose articles that involve plant extracts or compounds derived from plants including herbs and vegetables. If a compound or a plant extract has been studied by many authors, we selected only the most recent studies, as these studies likely cited previous published articles on the same compound or extract.

## 3. Major Molecular Mechanisms of Cisplatin-Induced Kidney Injury

As mentioned above, the major site of cisplatin accumulation in the nephron is the proximal tubules [24,25,26]. The entry of cisplatin into the tubular epithelial cells is thought to be mediated by two receptors: Crt1 and OCT2 (copper transporter 1 and organic cation transporter 2) [17]. Once inside the cells, cisplatin can rapidly accumulate in the mitochondria and damage mitochondrial components such as metabolic enzymes and mitochondrial DNA (mtDNA) [20]. It has been reported that mtDNA cisplatin adducts are more abundant than that of nuclear DNA [20], demonstrating that mitochondria are the major site of intracellular cisplatin accumulation. This would also indicate that mitochondria are the major organelle that receives cisplatin attack. As damage to mitochondria often culminates in an elevated level of ROS production, mitochondrial oxidative stress has been thought to be a major underlying mechanism of cisplatin-induced kidney injury [20,21,27].

In addition to the oxidative stress implicated in the pathogenesis of cisplatin kidney injury [28,29,30,31,32,33], inflammation and renal fibrosis have also been postulated to be involved in cisplatin-induced kidney injury [34,35,36,37,38,39]. Cisplatin-induced kidney injury may also involve NAD^+^ redox signaling pathways such as sirt3 [40], poly-ADP ribosylase (PARP) [41], and mitochondrial dynamics [18] including mitochondrial fission and fusion [42,43,44]. The eventual outcome of these cisplatin-impaired pathways converges toward cell death, renal fibrosis, and functional decline of the kidney [45]. Figure 2 summarizes the major molecular mechanisms that underlie cisplatin-induced kidney injury.

## 4. Rodent Models of Cisplatin-Induced Kidney Injury

When it comes to animal models, mice and rats are the widely used species for investigating the mechanisms of cisplatin kidney injury and evaluating the antioxidant properties of a variety of natural products [46,47,48,49]. Both AKI and CKD models can be created, depending on the objective of the studies, and there are no standard operation procedures that can be followed [48,49]. A general scheme for the use of mice and rats as cisplatin-induced kidney injury models is shown in Figure 3. For AKI, a single high dose of cisplatin is often applied to either the mouse or the rat. For CKD, multiple low dose cisplatin administration is often used and the treatment frequency and duration can also vary depending on the purpose of the studies. It has been reported that weekly intraperitoneal injection of low-dose cisplatin for 4 weeks can create a robust model of CKD [50,51]. However, the end point analysis of kidney function and measurement of kidney functional parameters after low-dose cisplatin induction of CKD can range from weeks to months [51,52]. It should be pointed out that many cisplatin studies using rodent models only involve healthy animals, instead of animals bearing cancers or tumors. Therefore, data obtained from healthy animals may not be comparable to those from cancer animals, given that cancer itself can affect kidney function and the vulnerability of a kidney to cisplatin toxicity [49]. Additionally, differences in genetic background of mouse strains can also affect the susceptibility of the kidneys to cisplatin challenges [53]. For example, the C57BL/J mouse lacks nicotinamide nucleotide transhydrogenase (NNT) [54,55,56,57,58,59] and should be used with extreme caution when the focus is studying NAD^+^ redox biochemistry in kidney disease.

## 5. Effects of Cisplatin on Major Individual NAD^+^-Dependent Redox Enzymes

Targeting NAD^+^ redox balance has been suggested as a strategy for fighting cisplatin-induced kidney injury [18,60]. Therefore, there has been an increasing interest in studying redox biochemistry and NAD^+^ redox signaling in the pathogenesis of cisplatin renal toxicity [32,61,62,63,64]. The major NAD^+^-dependent redox enzymes that may be involved in cisplatin-induced kidney injury are shown in Figure 4. These include mitochondrial complex I [65,66], sirtuins [40,67,68], alpha-keto acid dehydrogenases involving dihydrolipoamide dehydrogenase [69], NAD kinase (NADK) [70,71,72,73], CD38 [74,75,76], poly-ADP ribosylase [77,78,79], and nicotinamide nucleotide transhydrogenase (NNT) [57,80,81]. Numerous natural products that possess antioxidant activities have been shown to display antioxidant properties such as the inhibition of lipid peroxidation, DNA damage, and protein oxidation, which are collectively the popular parameters used to assess oxidative stress and antioxidant natural products [69,82,83,84]. However, many investigations did not analyze further to pinpoint the NAD^+^-implicated molecular mechanisms of natural plant products that are being tested. For example, with respect to NAD^+^-dependent oxidative stress and amelioration, the determination of the exact NAD^+^-involved redox enzymes that are involved or are modulated by the tested natural products has largely been unaddressed. Another example is mitochondrial complex I. Although it has been reported that complex I-generated superoxide anion is involved in cisplatin-induced kidney injury [19,85,86], exactly which subunits of complex I are responsible for the eventual superoxide production upon cisplatin stimulation have not been explored. In particular, future studies should focus on dissecting these potential NAD^+^-dependent redox enzymes involved in cisplatin-induced kidney injury. Nevertheless, a limited number of studies have shed light on these NAD^+^-dependent redox enzymes. For example, it has been reported that CD38 can mediate calcium mobilization in cisplatin-induced kidney injury [87]. Likewise, poly-ADP ribosylase has been found to be activated in cisplatin-induced kidney injury [88] and enhancement of sirtuin protein function can attenuate cisplatin-induced kidney injury [89]. Of note, studies on the potential modulation of NNT, complex I, NADK, and alpha keto acid dehydrogenase by plant-derived natural products are extremely lacking. Future studies may also need to be conducted on other cellular components, such as electron transport chain components complex II to IV, mitochondrial dynamics and biogenesis, and TCA cycle components as well as the fatty acid oxidation pathways. Elucidating the mechanisms of cisplatin damage to these components not only can explain the mechanisms of action of cisplatin, but can also provide novel insights into further strategies designed to counteract cisplatin kidney toxicity. It is conceivable that damage to the redox enzymes shown in Figure 4 would impair NAD^+^-associated redox balance, thereby accentuating kidney injury by cisplatin. Moreover, it is also conceivable that approaches elevating NAD^+^ content may lead to therapies [18,60,90].

It should be noted that among the major enzymes shown in Figure 4, poly-ADP ribosylase (PARP), in particular, PARP1, has been demonstrated to exert regulatory effects on the expression of many inflammatory proteins including IL-1-beta, TNF-alpha, IL-6, and toll-like receptor 4 (TLR4) [91]. PARP1 is upstream of TLR4, as pharmacological inhibition of PARP1 can attenuate the deleterious effect of cisplatin-induced inflammation in the kidney [92]. Likewise, knockout of TLR4 is protective against cisplatin-induced kidney injury [93,94] indicating that TLR4, among other TLRs [95], is required for cisplatin-induced renal toxicity. The downstream pathways of TLR4 such as the JNK and p38 pathways are likely involved in TLR4 knockout nephroprotection, as activation of each pathway by cisplatin was mitigated in the TLR4 knockout animals [91]. Nonetheless, how this signaling cascade from PARP1 to TLR4 to JNK and p38 is implicated in NAD^+^/NADH redox imbalance involved in cisplatin-induced kidney injury remains to be investigated in detail. Additionally, whether the other enzymes in Figure 4 could have a similar signaling role to that of PARP1 in regulating TLRs-mediated inflammation response in cisplatin-induced kidney injury also remain to be studied.

## 6. Counteracting Effects of Natural Products Derived from Plants

As shown in Table 1, where references are also provided, numerous natural products derived from plants, whether as a purified single compound or in an extract, have been tested for their counteracting effects on cisplatin-induced kidney injury. The general mechanisms of these natural products are summarized in Figure 5. These include blockage of cisplatin renal uptake and transportation [22], inhibition of oxidative stress [96,97], inhibition of inflammation [98,99], inhibition of P53 signaling pathways [52], inhibition of mitogen-activated protein kinases [100], attenuation of cell death, and enhancement of cellular antioxidant defense systems such as SOD, catalase, and the Nrf2 pathway [101,102]. Autophagy and mitophagy are also involved in cisplatin-induced kidney injury [51,103] and can be modulated by natural products for protective purposes [104,105,106,107]. It should be pointed out that administration of these natural products can be achieved either before cisplatin ingestion or after cisplatin ingestion, reflecting heterogeneous approaches to evaluating the ameliorating effects of a given natural product on cisplatin-induced kidney injury [49].

Among the numerous nephroprotective mechanisms shown in Figure 5, one particular mechanism needs to be highlighted: the Nrf2 signaling pathway [195,196,197]. It has been demonstrated that cisplatin induces the downregulation of the Nrf2 signaling pathway, leading to downregulation of Nrf2 target genes such as HO-1 and NQO1, the two major molecules executing Nrf2′s cytoprotective effects via anti-oxidation and anti-apoptosis [198]. Many studies have shown that cisplatin-induced downregulation of Nrf2 [199,200] can be reversed by natural products derived from plants [201,202,203]. Under normal conditions, Nrf2 is kept inactive in the cytosol through Keap 1 binding [204,205,206,207]. Upon exogenous stress stimulation, Keap 1 releases Nrf2, so that Nrf2 is able to translocate to the nucleus, where it binds to the antioxidant response element (ARP) to induce the expression of many cytoprotective molecules including HO-1 and NQO1 [204,205,206,207]. Accordingly, as shown in Table 1, numerous natural products have been demonstrated to be able to stimulate the release of Nrf2 from the Nrf2–Keap 1 complex, leading to an increased Nrf2 content in the nucleus and enhanced expression of cytoprotective molecules. Further evidence that supports these findings is that an Nrf2 knockout abolishes the nephroprotective effects of a given natural product [208]. It should be pointed out that while the downstream gene expression is well elucidated, the upstream events in the cytosol are less clear and remain to be defined. For example, how a natural product stimulates the release of Nrf2 from the Nrf2–Keap 1 complex is yet to be studied in detail.

## 7. Other Factors That Can Modulate Cisplatin-Induced Kidney Injury

It is worth noting that in addition to the natural products shown in Table 1, other approaches have been tested to counteract cisplatin-induced injury. These approaches include caloric restriction [209,210] and ketone body ingestion [211]. Both have been demonstrated to ameliorate and modulate cisplatin-induced kidney injury [209,210,211]. These approaches may be applied together with prescribed therapies to enhance the efficacy of given drugs for cisplatin-induced nephrotoxicity. It should also be noted that aging and obesity are prominent risk factors in cisplatin-induced renal toxicity [212,213,214] and such risk factors can be modulated by NAD^+^ precursors [215] and natural products [216,217].

## 8. Summary

While the pathogenesis of cisplatin-induced kidney injury is complex, the major underlying mechanisms converge on oxidative stress, inflammation, and renal fibrosis [11,17], which may involve major NAD^+^-dependent redox enzymes, such as mitochondrial complex I, CD38, NNT, NADK, PARP, alpha-keto acid dehydrogenases, and sirtuins. We also showed that numerous natural products tabulated in this review may directly or indirectly exert their renoprotective effects on cisplatin kidney toxicity via these NAD^+^ redox enzymes. A further understanding of the molecular mechanisms underlying cisplatin kidney toxicity may provide insights into design of novel strategies for counteracting cisplatin renal toxicity and increasing the clinical applications of cisplatin in cancer patients. In this context, animal models of cisplatin-induced kidney injury will continue to serve as invaluable tools.

## Figures and Tables

**Figure 1 biomolecules-12-01078-f001:**
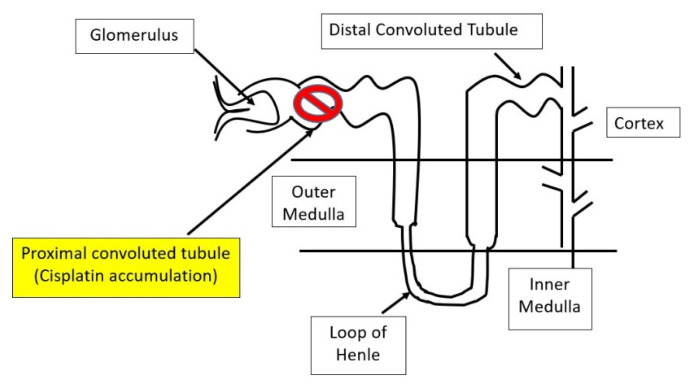
Diagram showing the proximal convoluted tubule (PCT) as the major site of cisplatin accumulation and toxicity in the nephrons.

**Figure 2 biomolecules-12-01078-f002:**
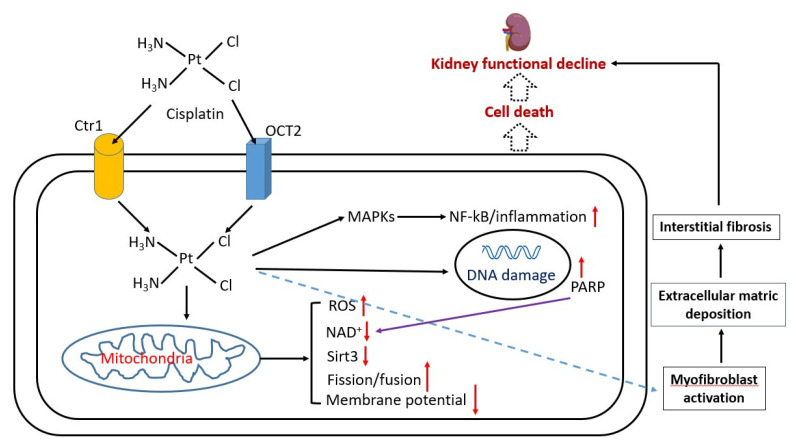
Major pathological mechanisms of cisplatin-induced kidney injury. Cisplatin enters into cell via copper transporter 1 (Ctr1) or organic anion transporter 2 (OCT2) receptors on the cell surface. Once inside the cell, cisplatin can go on to elicit a variety of actions or cellular responses such as nuclear and mitochondrial DNA damage, perturbation of mitochondrial function that can elevate ROS production, and decrease in NAD content and decrease in activity of NAD-dependent enzymes such as sirtuins. DNA damage could activate PARP, which consumes NAD, and in turn could further lower the NAD content, leading to NAD redox imbalance. Cisplatin can also activate inflammation-signaling pathways such as NF-kB activation via MAPKs. These events can result in interstitial fibrosis and eventual kidney failure.

**Figure 3 biomolecules-12-01078-f003:**
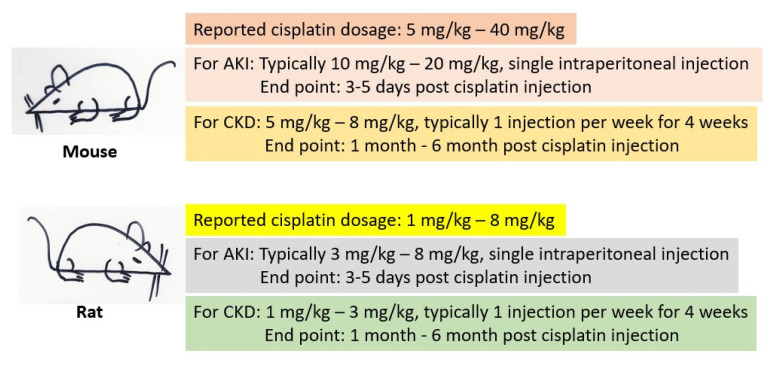
Outlines of rodent models used for studying cisplatin-induced kidney injury. Shown are the dose ranges for either mice or rats involving either AKI or CKD. It should be noted that these are just general guidelines for designing an experiment and should be modified for specific experimental objectives if needed.

**Figure 4 biomolecules-12-01078-f004:**
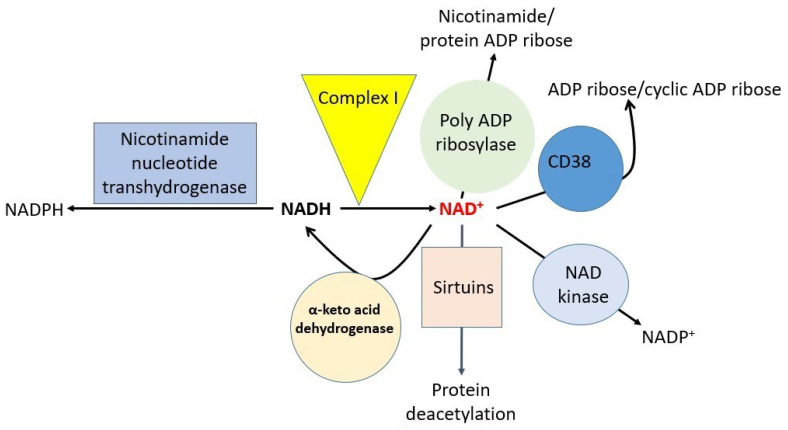
Major NAD-dependent redox enzymes that are potentially involved in cisplatin-induced kidney toxicity.

**Figure 5 biomolecules-12-01078-f005:**
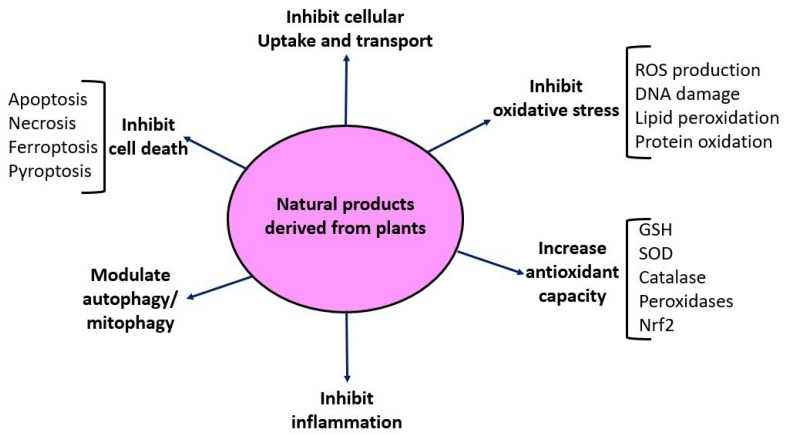
Schematic diagram depicting the protective mechanisms of natural products against cisplatin-induced kidney toxicity listed in Table 1.

**Table 1 biomolecules-12-01078-t001:** Counteracting effects of plant-derived natural products on cisplatin- induced renal toxicity *.

Natural Product	Rodent Model	Mechanism	Reference
4-hydroxyhalcone	HEK293 cell	Inhibiting ROS production	[108]
6-shogaol	Mouse	Anti-oxidative stress	[109]
10-dehydrogingerdione	Rat	Increasing GSH content	[110]
Acacia hydaspica R. Parker	Rat	Anti-oxidative damage	[111]
Alhagi camelorum	Rat	Increasing antioxidant activities	[112]
Andrographis Paniculata	Rat	Nrf2 signaling	[113]
Artemisia asiatica	LLC-PK1 cells	Anti-oxidative stress	[114]
Benzyl Isothiocyanate	Mouse	Anti-oxidative stress	[115]
Berberis integerrima	Rat	Anti-oxidative damage	[116]
Bisabolol	Mouse	Mitigating oxidative stress	[117]
Black bean extract	Rat	Anti-oxidation and anti-inflammation	[118]
β-lapachone	Mouse	Increasing NAD levels	[60]
Carrichtera annua DC	Rat	Anti-oxidative stress	[119]
Carvacrol	Rat	Anti-oxidative stress	[120]
Catapol	Mouse	Anti-oxidative stress	[121]
Chalcone	Mouse	Inhibiting necroptosis	[122]
Citrullus colocynthis Linn	Rat	Anti-oxidative stress	[123]
Citrus aurantium	Rat	Not determined	[124]
Clinacanthus nutans	NRK-52E cells	various protective effects	[125]
Coleus amboinicus extract	Rat	Increasing TGF-1β	[126]
Curcumin	Mouse	Anti-inflammation	[127]
Coumarins	Mouse	Suppressing renal inflammation	[128]
D-allose	Mouse	Suppressing renal inflammation	[129]
Daidzein	Mouse	Anti-oxidative stress	[130]
Danshen	Mouse	Nrf2 signaling	[131]
Daucus carota	Rat	Not determined	[132]
Dendropanoxide	Rat	AMPK/mTOR pathway	[133]
Dioscin	Rat/Mouse	Maintaining redox balance	[89]
Emodin	Rat tubular cells	Activating autophagy	[134]
Ephedra alata extract	Mouse	Reducing oxidative stress	[135]
Exacum lawii extract	Rat	Anti-oxidative damage	[136]
Ficus carica L. leaves	Rat	Anti-oxidative stress	[137]
Filipendula ulmaria extract	Rat	Anti-oxidative stress	[138]
Formononetin	Rat	Activation of Nrf2 pathway	[139]
Forskolin	Rat	Anti-oxidation and anti-inflammation	[140]
Galangin	Mouse	Attenuating oxidative stress	[141]
Ganoderma lucidum	Mouse/rat	Antioxidation	[142]
Garlic extract	Rat	Anti-oxidative stress	[143]
Genistein	Mouse	Decreasing ROS production	[144]
Ginkgo biloba	Rat	Inhibiting renal fibrosis	[145]
Ginsenoside Rg3	Mouse	Attenuating apoptosis	[146]
Green coffee beans extract	Mouse	Not determined	[147]
Huaier polysaccharide	Mouse	Anti-oxidative stress	[148]
Leea asiatica leaves	Mouse	Inhibiting lipid peroxidation	[149]
Honokiol	Mouse	Inhibiting mitochondrial fission	[150]
Licorice	HK-2 cells	Scavenging ROS	[151]
Liquiritigenin	Mouse	Nrf2/Sirt3 signaling pathways	[152]
Iosliquiritigenin	LLC-PK1 cells	Anti-oxidative stress	[153]
Jatropha mollissima extract	Rat	Anti-oxidative stress	[154]
Kahweol	Mouse	Suppressing inflammation	[155]
Maitake beta-glucan	Mouce	Anti-apoptosis	[156]
Matrine	Mouse	SIT3/OPA1 pathway	[157]
Momordica dioica Roxb.	Mouse	Anti-oxidative damage	[158]
Morus alba L extract	Rat	Anti-oxidative stress	[159]
Nigella sativa seed extract	Rat	Anti-oxidative damage	[160]
Opuntia ficus indica	Mouse	Anti-oxidative stress	[161]
Pleurotus cornucopiae	LLC-PK1 cells	Not determined	[162]
Plumbago zeylanica L	Mouse	Anti-oxidative stress	[163]
Polydatin	Mouse	Anti-oxidative stress	[164]
Polysulfide	Mouse	Anti-inflammation	[165]
Pomegranate rind extract	Rat	Anti-apoptosis	[166]
Puerarin	Rat	Upregulating microRNA-31	[167]
Punicalagin	Rat	Anti-oxidative stress	[168]
R. vesicarius L extract	Mouse	Anti-oxidative stress	[169]
Red ginseng	Rat	Anti-lipid peroxidation	[170]
Resveratrol	Rat	Anti-oxidative damage	[171]
Rheum turkestanicum	Rat	Decreasing oxidative damage	[172]
Rhus tripartitum extract	Rat	Increasing antioxidant potential	[173]
Ribes diacanthum Pall	Mouse	Enhancing antioxidant potential	[174]
Rutin	Rat	Anti-oxidative stress	[175]
Safflower seed extract	Mouse	Anti-oxidative stress	[176,177]
Sea lettuce extract	Rat	Anti-oxidative stress	[178]
Sesamin	Rat	Anti-oxidative stress	[179]
Seihaito (TJ-90)	Mouse	Anti-oxidative stress	[180]
Sonchus cornutus	Mouse	Anti-oxidative stress	[181]
Sorghum straw dye	Rat	Anti-oxidative stress	[182]
Stachys pilifera benth	Rat	Anti-oxidative damage	[183]
Stevia	Mouse	Anti-oxidative stress	[184]
Sulforaphane analogues	LLC-PK1 cells	Anti-apoptosis	[185]
Vietnamese Ginseng	LLC-PK1 cells	Improving kidney function	[186]
Tanshinone I	Mouse	Increasing antioxidant enzymes	[187]
Terminalia chebula	Rat	Anti-apoptosis	[188]
Tetrahydrocurcumin	Rat	Decreasing oxidative damage	[189]
Troxerutin	Rat	PI3K/AKT pathway	[190]
Tukhm-e-karafs	Rat	Reducing ROS production	[191]
Whortleberry	Rat	Antioxidation	[192]
WIthania coagulans extract	Rat	Anti-oxidative stress	[193]
Zingerone	Rat	Inhibiting oxidative stress	[194]

* Please note that this table is not meant to be exhaustive. Rather, we chose representative natural plant products reported recently in the literature.

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
