# Peer review of "Cisplatin-Induced Kidney Toxicity: Potential Roles of Major NAD+-Dependent Enzymes and Plant-Derived Natural Products"

_biomolecules, 2022, doi:10.3390/biom12081078_

Round 1

Reviewer 1 Report

This is an interesting manuscript; however, there are some points to be revised as follows:

1) As title mentioned, the authors had better describe "plants-derived natural products".

2) In order to strengthen the evidence, a reference (Int J Mol Sci. 2016 Nov 1;17(11):1826.) should be added after the description "As damage to mitochondria of ten culminate in an elevated level of ROS production, mitochondrial oxidative stress has been thought to be a major underlying mechanism of cisplatin-induced kidney injury" (Page , Lines 73-75)

3) Table 1: please line up at the begining of terms in Rodent model and Mechanism, and reference.

Reviewer 2 Report

in this manuscript, the authors described the underlying mechanisms of cisplatin kidney injury in the context of NAD+-dependent redox players including mitochondrial complex I, NAD kinase, CD38, 18 sirtuins, poly ADP ribosylase polymerase, and nicotinamide nucleotide transhydrogenase (NNT) and their potential contributing roles in amelioration of cisplatin-induced kidney injury conferred by natural products derived from plants.

 ·       However, in the current study, the value addition is minimal and therefore the manuscript lacks novelty. Here are some ideas to improve the outcome of the review article:

Ø  More discussion especially for figure 4 and

Ø  The authors need to explore the detailed pathway associated with TLRS (upstream and downstream)

Ø  The authors need to explain Nrf2/Keap1 expression in this study since the expression of both the genes is downregulated.

Ø  To explain the nuclear translocation of Nrf2 and explore the levels of keap1 in the cytosol before and after cisplatin treatment.

Ø  Also, downstream proteins, HO-1 and NQO1 need to be investigated.

Ø  The manuscript needs to be thoroughly checked for language, grammar, and typos.

 ·       The authors did not explain the methodology adopted to perform this review.

·       The figures are not professionally prepared

·       References are not in style and the font is incorrect 18 pages of references for only 5 pages of text is not reasonable

·        
